# Longitudinal outcomes of gender identity in children (LOGIC): a study protocol for a prospective longitudinal qualitative study of the experiences and well-being of families referred to the UK Gender Identity Development Service

Kathy McKay [1,2] Eilis Kennedy [2,3] Chloe Lane,[2] Talen Wright,[4] Bridget Young[1]

For numbered affiliations see end of article.

**Correspondence to**
Dr Kathy McKay;
kmckay@tavi-port.nhs.uk

## ABSTRACT

**Introduction** Gender Identity Development Services (GIDS) worldwide have experienced a significant increase in referrals in recent years. However, little is currently known about the experiences of the children and young people (CYP) and their families attending these services and the influences on their well-being. Most published qualitative studies have explored gender identity and gender questioning CYP from either a parental perspective or in an adolescent sample. Consequently, there is a need for research to explore the voices of younger children and adolescents who are referred to GIDS. This study aims to address current gaps in understanding of the experiences of CYP referred to the UK GIDS, specifically regarding the personal, familial, educational, and social contexts in which CYP and their parents/caregivers are navigating gender identity, social and physical transition, and the healthcare system.

**Methods and analysis** A prospective longitudinal qualitative study examining the experiences and well-being of CYP referred to the UK GIDS. A purposive sample of up to 40 families will be recruited. Families will be eligible for the study if the child or young person was aged 3–14 years at the time of referral to the GIDS. Semistructured interviews will be conducted with both the child/young person and their parents/caregivers. Analysis of anonymised interview transcripts will be interpretive and pluralistic, informed by both narrative and thematic approaches. This study forms part of a wider programme of research investigating longitudinal outcomes of gender identity in children (the LOGIC Study).

**Ethics and dissemination** The proposed study has been approved by the UK Health Research Authority and London–Hampstead Research Ethics Committee as application 19/LO/0857. The study findings will be published in peer-reviewed journals and presented at both conferences and stakeholder events.

## INTRODUCTION

Gender Identity Development Services (GIDS) are specialist services for children and young people (CYP) presenting with

### Strengths and limitations of this study

► This prospective study will provide longitudinal qualitative data on the experiences and well-being of children, young people and their families referred to the UK Gender Identity Development Service (GIDS) between the ages of 3 and 14 years, over a 2-year period.

► Semistructured interviews will be conducted with both children and young people (CYP) who are referred to the service and their parents/caregivers in order to explore both perspectives.

► The sample will include both CYP who are referred to the service and continue to access treatment, as well as those who are referred and subsequently decide not to progress in the service.

► Findings from the study will be used to inform treatment pathways for CYP presenting to GIDS.

► As this is a 2-year prospective study, participants' experiences and well-being beyond this period, and their physical/medical treatment choices, will not be ascertained, although we intend to explore further funding opportunities that would enable us to extend the data collection and participant follow-up beyond the initial 2 years.

questions and challenges in relation to their gender identity development. Young people attending such services may or may not experience gender dysphoria. Gender dysphoria is defined as distress arising as a result of incongruence between experienced gender identity and the gender assigned at birth.[1] In the UK, referrals to the national GIDS have significantly increased over the past few years from 678 referrals in 2014/2015 to 2728 referrals in 2019/2020.[2] However, little is currently known about the experiences over time of

the CYP and their families attending GIDS or other such services.[3]

Most qualitative studies undertaken to date have explored gender identity and gender questioning CYP's experiences from a parental perspective. For example, two studies have examined parental experiences of their child socially transitioning (ie, changing clothing, hairstyle, first name and pronouns to better reflect their gender identity).[4 5] Specifically, findings from these studies suggested that early social transition may have benefits for CYP in relation to mood and family and social relationships. Parental perceptions of emotional and behavioural difficulties in their pre-pubertal transgender and gender non-conforming children have also been explored, and this work has indicated the need to adapt existing interventions to better support these children.[6] Findings from these studies are informative in terms of understanding parental perspectives on their children, but it is important to also consider the experiences of CYP directly, particularly in relation to identifying how services can be tailored to best support the needs of CYP and their families.

A recent study examined both transgender CYP and parent/caregiver perspectives in relation to imagining the future.[7] This study included 16 CYP, aged 7–18 years, as well as 29 caregivers, and transcripts from both groups were analysed together. This study reported on CYP's views and experiences of comparing themselves with others, gender-affirming hormones, gender-affirming surgery, gender norms, questioning their transgender experience, expectations for romantic relationships, uncertainty about the future, and worries about physical and emotional safety. Findings from this study provide insights into how both CYP and their parent/caregiver perceive the future. Although the sample included younger children, the study focused very specifically on one topic that is imagining the future, and it will therefore be important for further research to consider broader experiences of CYP and their parents/caregivers from the perspective of both parties.

Few other qualitative studies have explored the voices of CYP directly, particularly in relation to younger, pre-pubertal CYP. Studies have predominantly focused on adolescents. For example, research has investigated adolescents' opinions of specific topics such as the use of puberty suppression.[8] Furthermore, a study of adolescents aged 14–18 years explored experiences of gender identity using semistructured biographical interviews.[9] Participants in this study reported that the period between 10 and 13 years was crucial in relation to their developmental trajectory. In addition, this study reported that increases or decreases in gender dysphoria were influenced by changing social environments, anticipated and actual bodily changes occurring during puberty and the emergence of sexual and romantic feelings. Although this study included interviews conducted directly with young people, they were asked to describe their gender identity retrospectively and recall experiences from their childhood. Thus, it is important to add further to the understanding of developmental trajectories and the factors influencing these by directly interviewing a younger sample of CYP and using a prospective longitudinal approach to identify changes over time, as they occur.

A further recent study explored priorities and experiences of adolescents with diagnoses of both gender dysphoria and autism spectrum disorder (ASD) over a 22-month period, with semistructured interviews conducted at three time points.[10] This study tracked participant experiences over time, enabling diverging developmental trajectories to be identified. At the end of the study, some participants were affirmed in their gender identity across all settings and some showed attenuating gender dysphoria, despite all participants meeting criteria for gender dysphoria initially. Key themes identified were: urgent gender needs, the complexity of being both gender diverse and autistic, the realisation of gender identity over time, the impact of bias and harassment on gender expression and overall confidence in the future. Although this study used a prospective longitudinal approach, it focused on a specific sample of gender dysphoric CYP with co-occurring ASD. It will therefore be important to extend the findings from this study to a wider, more diverse sample in order to broaden understanding of the priorities and experiences of CYP with gender dysphoria.

This study also seeks to extend on the work undertaken by Pullen Sansfaçon and colleagues in Canada.[11 12] While not longitudinal, their findings on the experiences of gender diverse and trans CYP, particularly in terms of resilience and access to healthcare and the ways in which these are interwoven, highlight the importance of hearing the voices of those with lived experience. Support for CYP needs to not only encompass their physical and mental healthcare, but also ensure they are supported through their family and social connections.

In summary, this study aims to address current gaps in understanding of the experiences and well-being of CYP referred to the UK GIDS, specifically in relation to the personal, familial, educational, and social contexts in which CYP and their parent/caregiver are navigating gender identity, social and physical transition, and the healthcare system. Of note, the vast majority of qualitative studies have been conducted outside of the UK and thus it is important to explore experiences of CYP accessing services within a UK setting. CYP and their parent/caregiver will be followed over time, allowing changes to be tracked. The types of supports and interventions that are needed and how access to these can be improved will be examined. This study is particularly novel and innovative in terms of using a prospective design, recruiting families currently on the waitlist for the GIDS at baseline and using a longitudinal approach to examine changes over time. Furthermore, this study will include pre-pubertal and early pubertal CYP, a group which are currently understudied.

## Aims

This study aims to explore the experiences and well-being of a diverse sample of CYP referred to the UK GIDS, and their parent/caregiver, over a 2-year period. Specifically, the study aims to explore and examine (1) CYP experiences of gender identity, how this changes over time and links to health and well-being; (2) parental experiences of having a gender-questioning child, how this changes over time and support needed to promote their child's health and well-being; (3) CYP and parental perspectives on outcomes that are important for evaluating health services and treatments for gender-diverse children; (4) CYP and parental experiences of health services, therapies and support and how these can be enhanced.

## METHODS AND ANALYSIS
### Study design

A prospective longitudinal qualitative study examining the experiences and outcomes of CYP referred to the UK GIDS. This study will subsequently be referred to as LOGIC-Q and forms part of a wider programme of research investigating the Longitudinal Outcomes of Gender Identity in Children (the LOGIC Study).[13] This programme of research uses a mixed-methods approach, incorporating both quantitative and qualitative longitudinal studies to investigate experiences and outcomes of families referred to the UK GIDS.

### Participant identification and recruitment

Families in which the CYP were aged 3–14 years at the time of referral to the GIDS and who are currently on the waitlist for their first appointment will be invited to participate in the LOGIC Study (see Protocol for LOGIC cohort study for further details). During the consent process for the LOGIC quantitative study, parents/caregivers will be asked whether they agree to being contacted about participating in LOGIC-Q. Following completion of the LOGIC quantitative study, parents/caregivers who provide consent and meet the sampling criteria will be invited to participate in LOGIC-Q and will receive age-appropriate information sheets about the study. Interviews will be arranged with families who subsequently agree to participate in the study.

### Sampling

A purposively sampled subset of up to 40 families will be recruited. The sample will be diverse in relation to assigned sex at birth, current age, social transition status (socially transitioned, partially socially transitioned and no social transition), ethnicity, diagnosis of ASD, socio-economic status and geographical location. These details will be included in the quantitative data accessible to a qualitative researcher (KM) who will then, in discussion with the rest of the qualitative team, make contact with those families who have both consented to the contact but also fit to the agreed on quotas to ensure as diverse a sample as possible. In addition, sampling will seek to include CYP who do not progress to seek medical intervention, as well as those who do seek medical intervention, but this may not necessarily be known until the end of the study. Overall, sampling to the baseline interviews will cease when saturation has been reached.[14] Saturation will be seen to have been reached when, after discussion among the qualitative team, it is felt that there are no more experiential themes identified within the narratives.

### Data collection

Where possible, participants will be interviewed face-to-face either in their homes or in a setting of their choice. In instances when travel is not an option, interviews will be conducted via a video call or the telephone. In light of COVID-19 and the travel and social distancing restrictions that have been in place in the UK since March 2020, all interviews will be conducted via video call or telephone. In instances when travel is not an option, interviews will be conducted via a video call or the telephone. As restrictions begin to ease, steps to monitor and ensure the safety of the families and interviewer will take precedence. In-person interviews will be undertaken on a case-by-case basis, where both the family and interviewer are in agreement, and travel and social contact is both safe and allowed.

All participants will complete an initial interview and will then be interviewed again 12 and 24 months later; in total, families will be asked to undertake three interviews over the course of the study. Interviews will be conducted by an experienced qualitative researcher (KM). In order to maintain confidentiality and enable both CYP and their parent/caregiver to speak freely about their experiences, interviews will ideally be conducted separately. However, should families wish to be interviewed together, this will be accommodated. In the event that a participant becomes distressed or discloses a risk/safeguarding concern during an interview, the researcher will adhere to the study risk protocol.

Semistructured interviews will be conversational and topic-guided. Topic guides will be developed in collaboration with the peer researcher, wider project team, the GIDS and in consultation with external patient and public involvement groups. For CYP, questions will cover the following topics: relationships; appearance and feelings about self; health services, support and decisions; and school. Depending on the age of the participant, questions will be adapted appropriately. For parents/caregivers, questions will cover the following topics: child's first questions/exploration about gender identity; child's current expression of gender identity and well-being; responses of family and others to child; child's needs and support from services to address child's needs; reflections on gender identity; outcomes; looking ahead. Towards the end of the interview, participants will also be given the opportunity to discuss any other topics that have not yet been mentioned. During the course of the study, topic guides will be developed to include unanticipated and emerging issues, as well as avoid redundancy, as informed

by ongoing analysis. Redundancy will also be avoided as three interviews over 3 years cover a time period in CYP's life where there are myriad physical, social and emotional changes, including puberty, school and accessing healthcare. For interviews that need to be done via video call, to help support the engagement of children, particularly for interviews at the second and third time points, they will also be offered the option to draw a picture of themselves and their family demonstrating how they are feeling. While this will not be used for data collection per se, the drawing may help a child feel more comfortable in the space and create a starting point for talking.

In order to create a 'through line' of narrative in this longitudinal study, subsequent interviews with each family will include a set of core questions from the original topic guide, as well as questions based on previous answers. This enables changes in different experiences over time to be mapped and also allows participants to look back in time (to what has happened) and forward (to what they hope for).[15] In addition, it provides an opportunity for participants to reflect on how life has changed through both rigid time frames, like birthdays and school years, and more fluid time frames, like changes in maturity or self-reflection. Following the interview, the researcher will write field notes to allow for self-questioning and reflexive practice, as well as to help provide context to the interview transcript.[16 17] This will also inform subsequent interviews and allow bespoke prompts to be developed, enabling a participant-centred approach.

## Analysis

The NVivo V.12 software package will be used to support data management and analysis. Interviews will be audio-recorded using digital recorders with encryption facilities and transcribed by a professional transcription agency. Analysis of anonymised interview transcripts will be interpretive and pluralistic, informed by both narrative and thematic approaches.[18–21] Narrative inquiry is suited to a longitudinal qualitative study design and the aim of mapping change over time as it allows the analysis to be undertaken in multiple directions—inward (changes in and perceptions of self), outward (changes in and perceptions of others and the world), backward (past) and forward (future).[22] A narrative inquiry framework also grounds the analysis in the lived experiences of the participating parents and children.[23] Procedurally, analysis will draw on the framework method, a widely used approach that is suited to working with large qualitative datasets and facilitating the involvement of multi-disciplinary teams in the analysis.

The analytical process will be grounded in the guidelines around research integrity and fidelity as outlined by Levitt and colleagues.[24] In practical terms, KM will read and reread the transcripts several times, including listening to the audio alongside, while also taking notes, to create thick descriptions of experiences. The field notes attached to each interview transcript will also be considered in a contextual sense during this process. These will be read and discussed among the qualitative team—KM, BY, EK and TW. Inductive themes

will explored by KM and refined with the qualitative team. Themes will also be checked with the participant advisory group. While we will initially analyse year 1 data thematically, as year 2 and 3 data become available, we will increasingly turn to narrative analysis. This will focus on how participants order, sequence and present their stories and 'turning points' in their experiences of services, their relationships and understanding of themselves.

Analysis will explore across themes and cases at particular points in time as well as within cases and themes over time.[15 21] Initially, data from CYP and parents will be analysed separately in order to ensure that each perspective is given equal 'weight', before comparing within families to identify commonalities and divergences between CYP and parents/caregivers. As data from the follow-up interviews become available, continuities and changes over time will be examined. In addition, data from the larger quantitative study will be used to illuminate the analysis.[25] Given the paucity of previous qualitative research with CYP, there is little to inform selection of relevant theory at the outset of our study. However, as the analysis progresses, we will continue to look for theory that could illuminate interpretations of the data with the key aim of informing health services for children, young people and their families. As the study progresses, meetings will be held with members of the wider research team to review transcripts and discuss how the interviews have been progressing, develop the topic guide for follow-up interviews and to develop the analysis. The research team will also monitor whether saturation has been reached.

## Reflexivity

In order to further ensure methodological rigour and transparency, it is important to reflect on the positioning of the qualitative team in relation to the study, and how that may affect the data gleaned. KM is an experienced qualitative researcher who has worked extensively in sensitive areas of mental health research. TW is a lived experience co-investigator whose current PhD intends to examine how social determinants impact on the mental health of trans and gender-diverse people. EK is an experienced child and adolescent psychiatrist and clinical researcher. BY has extensive experience as a qualitative researcher focusing on psychological processes in healthcare and clinical research.

## Patient and public involvement

The LOGIC Study was developed in collaboration with a service user co-applicant (TW), UK GIDS users and external stakeholders. CYP and their families using the UK GIDS were consulted regarding their priorities for research at GID Service User Family Days in London and Leeds. This informed the research proposal and application for funding which was shared and further reviewed with parents of young people accessing the service. Throughout the study, the research team will host CYP and parent/caregiver advisory groups who will meet every 6 months to provide ongoing feedback and advice on the study. These will include families who are participating in the LOGIC Study.

## ETHICS AND DISSEMINATION

This study has been approved by the Health Research Authority and London–Hampstead Research Ethics Committee (19/LO/0857). LOGIC-Q will draw significantly on the recommendations set out by Vincent[26] which are specifically focused on the safety and agency of gender-diverse participants in research. The LOGIC-Q team are also guided by the Nuffield Guidance[27] in working with parents and children in research studies, especially ensuring that CYP have agency in their research participation and that the data collection methods are appropriate and flexible to their needs. The study findings will be published in peer-reviewed journals and presented at both conferences and stakeholder events.

### Informed consent

Families will be provided with information sheets and given the opportunity to ask the researcher questions before deciding whether or not to participate in the study. Before commencing the interview, the parent/caregiver will be asked to provide informed consent. All CYP will be asked to assent throughout the study. Acknowledging that there can be a potential power imbalance in research, KM will work hard to ensure that all the families understand why the research is being conducted, what their contribution will mean to the study, the value of their participation, and how their confidentiality and anonymity will be ensured and respected. KM will initially engage with the parents, answering any questions they have and ensuring that everyone is comfortable talking to her. Every participant will be informed that they have the right to withdraw from the study at any time, without providing a reason. As KM is in contact with the parent rather than the CYP prior to the interview, gaining the CYP's assent at the time of the interview is vital. This will be done by thoroughly explaining the research again, what doing an interview means, and about confidentiality and anonymity. For younger children particularly, this will also mean that KM will adapt the interview process to suit the child, whether this means asking questions of a favourite toy as well as the child, allowing the child to move in and out of the room and asking them questions as they reappear, and not pushing a child to answer a question they show discomfort with. It could even mean stopping the interview altogether if there are signs a child does not wish to continue, without expecting them to ask for that directly.

### Sensitive topics

In the first interview, the researcher will strive to establish rapport with families and make the interviews as enjoyable as possible. The first interview will also be used as a trust-building exercise with the families which is particularly important when children are involved. Interviews may cover sensitive topics and so the researcher will monitor participants throughout the interview and if a participant becomes distressed, the researcher will offer to temporarily stop or completely discontinue the interview. The researcher has significant experience in conducting interviews on sensitive topics. Participants will be debriefed at the end of the interview and the researcher will check how they are feeling.

### Confidentiality

Pseudonyms/identity codes will be held in a password-protected database on an encrypted National Health Service (NHS) server at the Tavistock and Portman NHS Foundation Trust. The names of people and/or places mentioned during the interviews will be removed from transcripts before analysis and storage. No identifiable data will feature in the dissemination of results.

### Data storage

All data will be held in a password-protected database on an encrypted NHS server at the Tavistock and Portman NHS Foundation Trust. Data processing agreements and data sharing agreements will be enforced between the Tavistock and Portman NHS Foundation Trust and study collaborators.

**Author affiliations**
[1]Public Health, Policy and Systems, Institute of Population Health, University of Liverpool, Liverpool, UK
[2]Research and Development Unit, Tavistock and Portman NHS Foundation Trust, London, UK
[3]Research Department of Clinical, Educational and Health Psychology, University College London, London, UK
[4]Division of Psychiatry, University College London, London, UK

**Contributors** Conceptualisation—EK, BY, KM and TW. Methodology—KM and BY. Formal analysis—KM. Validation—KM, TW, BY and EK. Writing (original draft preparation)—KM and CL. Writing (review and editing)—KM, EK, CL, TW and BY. Supervision—BY and EK. All authors have read and agreed to the published version of the manuscript.

**Funding** This work was supported by the National Institute for Health Research, Health Service and Delivery Research (grant number 17/51/19).

**Competing interests** None declared.

**Patient and public involvement** Patients and/or the public were involved in the design, or conduct, or reporting, or dissemination plans of this research. Refer to the Methods section for further details.

**Patient consent for publication** Not required.

**Provenance and peer review** Not commissioned; externally peer reviewed.

**ORCID iDs**
Kathy McKay http://orcid.org/0000-0002-5536-2522
Eilis Kennedy http://orcid.org/0000-0002-4162-4974

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
