## [Reviewer comments · BMJ Open]

ARTICLE DETAILS

TITLE (PROVISIONAL)	Longitudinal Outcomes of Gender Identity in Children (LOGIC): A study protocol for a prospective longitudinal qualitative study of the experiences and wellbeing of families referred to the UK gender identity development service
AUTHORS	McKay, Kathy; Kennedy, Eilis; Lane, Chloe; Wright, Talen; Young, Bridget

VERSION 1 – REVIEW

REVIEWER	O'Connor, Cliódhna University College Dublin School of Medicine and Medical Science, SMMS
REVIEW RETURNED	19-Mar-2021

GENERAL COMMENTS	Thank you for the opportunity to review this manuscript. This research project is highly necessary and I am delighted to see it in motion. The protocol seems considered and well-designed, and I would expect it to produce valuable original data. I have a few suggestions, queries and points of clarification that the authors might consider in developing the protocol. • I would appreciate more detail on the purposive sampling procedures. How will diversity of demographic/clinical characteristics be assessed – for example will quotas be used? The authors could also be more specific about how they will define and determine “saturation and adequate information power”.• Minimal detail is provided about the child assent procedures. What strategies will the research team use to ensure children’s assent is fully informed? What will happen if parents consent but the child refuses or expresses reluctance to participate?• I would encourage the researchers to consider offering the drawing opportunity in all interviews, not just the video-call interviews. In my own experience conducting interviews with children on sensitive topics, I have found that offering a paper and pencil can promote child engagement by reducing some of the ‘pressure’ of the unfamiliar interview situation e.g. related to eye contact, finding the right words etc. This would also promote consistency across all interview modalities. The possibility that the drawings may be used as data is briefly mentioned – if this is a concrete possibility, analytic approaches to this data should be elaborated.• The ‘longitudinal’ nature of the study is not entirely clear to me. For example, I do not see information on the number of interviews each family is expected to complete, the likely duration between interviews, or exactly how the successive interviews will build on each other and avoid redundancy.• In the Analysis section, I would expect to see more information on the approach to code/theme development, e.g. whether codes will
--

	be inductive/deductive/abductive, the strategies that will be used to develop codes into higher-order categories or themes. I would also like to see some information on how trustworthiness or quality criteria will be integrated into the analysis, e.g. formal intercoder reliability/comparison, member checking, thick description...  • It might be appropriate to include a section on Reflexivity where the researchers reflect on their own positioning in relation to the topic and participants, and how that may affect the data gleaned. • Have or will any pilot interviews be conducted? • “As this is a two-year prospective study, participants’ experiences and wellbeing beyond this period, and their physical/medical treatment choices, will not be ascertained.” – While I am not familiar with the scientific or logistical reasons behind this decision, it does seem a shame to rule out any possibility for further follow-up as the children age. I would encourage the authors to consider whether they might design their consent and data protection procedures so that some possibility for future follow-up is left open.
--	---

REVIEWER	Cariola, Laura The University of Edinburgh
REVIEW RETURNED	29-Mar-2021

GENERAL COMMENTS	This prospective study seeks to provide a longitudinal qualitative exploration with both children and young people and their families to better understand their experiences in relation to the UK GIDS. This is a valuable study and the authors discuss the contribution of this innovative qualitative study to the existing knowledge and understanding of the topic. Ethical issues have been clearly considered and discussed. Ethics codes of conduct/standards that guide this study could be mentioned. The relationship between the researcher and participants has been also discussed, and it would be useful to mention potential issues around power imbalance in the interview situation and how this would be addressed in the research. It would be also helpful to know how the researcher would address participants’ comments during the interviews/data collection and potential implications to make changes in the research design. In relation to Covid-19 risks to health in face-to-face data collection, it would be helpful to have some information about risk assessment. In relation to the analytic section, it is not clear how the analysis would be operationalised. The proposal of an interpretivist and pluralistic approach to thematic analysis and narrative inquiry is confusing. Arguably most qualitative approaches assume an interpretivist position. To improve the analysis section I would suggest providing a more clear outline of thematic analysis and how it will be used longitudinally from a narrative inquiry perspective. Currently, the analytic approach is unclear, for example, how the data will be analysed or coded, and how validity of the themes will be established. In case several researchers code the data, it would be important to explain how coding consistency will be established. In relation to the aims of the study, the notion of ‘describing’ (page 9, line 23) is not consistent with the qualitative approach (constructivist/interpretivist position) that preliminary seeks to explore.
--

	LOGIC-Q (page 9, line 48) requires disambiguation.
REVIEWER	Medico, Denise University of Quebec in Montreal, Sexology
REVIEW RETURNED	31-Mar-2021
GENERAL COMMENTS	Access to care for gender diverse children is an important issue, which has been punctuated by polarized perspectives in the scientific literature, and in the mass media. It is critical to develop empirical unbiased scientific data in light of the current debates on gender diverse children. The proposed research will provide researchers and health care providers alike with significant insight. The research objectives of the study are clear, and its design method is relevant. A literature review would not be complete without an appraisal of worldwide initiatives. A similar yet non-longitudinal study has been done in Canada by a nationwide team of researchers led by Professor Annie Pullen Sansfaçon. The results of the aforementioned studies have been published. Another international longitudinal multi-site study (CA, USA, CH, AUS, UK) will begin in the near future. See: 1. Pullen Sansfaçon, A., Suerich-Gulick, F., Temple-Newhook, J., Feder, S., Lawson, M., Ducharme, J., Ghosh, S., & Holmes, C. (2019). The experiences of gender diverse and trans children and youth considering and initiating medical interventions in Canadian gender-affirming specialty clinics. International Journal of Transgenderism, 20(4), 371-387. https://doi.org/10.1080/15532739.2019.1652129 2. Pullen Sansfaçon, A., Hébert, W., Lee, E. O. J., Faddoul, M., Tourki, D., & Bellot, C. (2018). Digging beneath the surface: Results from stage one of a qualitative analysis of factors influencing the well-being of trans youth in Quebec. International Journal of Transgenderism, 19(2), 184–202. https://doi.org/10.1080/15532739.2018.1446066

VERSION 1 – AUTHOR RESPONSE

Reviewer: 1

Dr. clíodhna oconnor, University College Dublin School of Medicine and Medical Science

Comments to the Author:

Thank you for the opportunity to review this manuscript. This research project is highly necessary and I am delighted to see it in motion. The protocol seems considered and well-designed, and I would expect it to produce valuable original data. I have a few suggestions, queries and points of clarification that the authors might consider in developing the protocol.

Thank you so much for your supportive comments about the paper.

- I would appreciate more detail on the purposive sampling procedures. How will diversity of demographic/clinical characteristics be assessed – for example will quotas be used? The authors could also be more specific about how they will define and determine “saturation and adequate information power”.

We apologise for the confusion and have added more detail about the sampling process, as well as saturation. We have removed information power as it added confusion and saturation is both appropriate and relevant to the process:

“These details will be included in the quantitative data accessible to a qualitative researcher (KM) who will then, in discussion with the rest of the qualitative team, make contact with those families who have both consented to the contact but also fit to the agreed upon quotas to ensure as diverse a sample as possible. In addition, sampling will seek to include CYP who do not progress to seek medical intervention, as well as those who do seek medical intervention, but this may not necessarily be known until the end of the study. Overall, sampling to the baseline interviews will cease when saturation has been reached (12). Saturation will be seen to have been reached when, after discussion amongst the qualitative team, it is felt that there are no more experiential themes identified within the narratives”.

- Minimal detail is provided about the child assent procedures. What strategies will the research team use to ensure children’s assent is fully informed? What will happen if parents consent but the child refuses or expresses reluctance to participate?

We agree that children’s assent is vital and have added more detail in the section on Informed Consent:

“As KM is in contact with the parent rather than the CYP prior to the interview, gaining the CYP’s assent at the time of the interview is vital. This will be done by thoroughly explaining the research again, what doing an interview means, and about confidentiality and anonymity. For younger children particularly, this will also mean that KM will adapt the interview process to suit the child, whether this means asking questions of a favourite toy as well as the child, allowing the child to move in and out of the room and asking them questions as they reappear, and not pushing a child to answer a question they show discomfort with. It could even mean stopping the interview altogether if there are signs a child does not wish to continue, without expecting them to ask for that directly”.

- I would encourage the researchers to consider offering the drawing opportunity in all interviews, not just the video-call interviews. In my own experience conducting interviews with children on sensitive topics, I have found that offering a paper and pencil can promote child engagement by reducing some of the ‘pressure’ of the unfamiliar interview situation e.g. related to eye contact, finding the right words etc. This would also promote consistency across all interview modalities. The possibility that the drawings may be used as data is briefly mentioned – if this is a concrete possibility, analytic approaches to this data should be elaborated.

We absolutely agree with drawing being a great way to reduce the potential pressure of an interview for a child, and we apologise for any confusion caused. We have reworked the section to demonstrate that the drawings will not be used as data:

“While this will not be used for data collection per se, the drawing may help a child feel more comfortable in the space and create a starting point for talking”. Further details on other ways the researcher will work to ensure every child feels comfortable in the interview space are included in the section on Informed Consent (see above and in the manuscript).

- The ‘longitudinal’ nature of the study is not entirely clear to me. For example, I do not see information on the number of interviews each family is expected to complete, the likely duration between interviews, or exactly how the successive interviews will build on each other and avoid redundancy.

Apologies for the lack of clarity. We have added the following sentences to the Data Collection section which we hope give more detail:

“All participants will complete an initial interview and will then be interviewed again 12 and 24 months later; in total, families will be asked to undertake three interviews over the course of the study”.

“During the course of the study, topic guides will be developed to include unanticipated issues, as well as avoid redundancy, as informed by ongoing analysis. Redundancy will also be avoided as three interviews over three years cover a time period in CYP’s life where there are myriad physical, social, and emotional changes, including puberty, school, and accessing healthcare”.

- In the Analysis section, I would expect to see more information on the approach to code/theme development, e.g. whether codes will be inductive/deductive/abductive, the strategies that will be used to develop codes into higher-order categories or themes. I would also like to see some information on how trustworthiness or quality criteria will be integrated into the analysis, e.g. formal intercoder reliability/comparison, member checking, thick description...

Thank you for these thoughtful considerations. A strong analytical process is very important to us, and we are very much guided by Levitt and colleagues (see addition to manuscript) where processes for research integrity are bound to the type of qualitative methodology and methods used. Our analytical process has several aspects explained here and in the Reflexivity section that ensure validity and rigor. We have added this further detail into the Analysis section:

“The analytical process will be grounded in the guidelines around research integrity and fidelity as outlined by Levitt and colleagues (24). In practical terms, KM will read and re-read the transcripts several times, including listening to the audio alongside, while also taking notes, to create thick descriptions of experiences. The field notes attached to each interview transcript will also be considered in a contextual sense during this process. These will be read and discussed amongst the qualitative team – KM, BY, EK, and TW. Inductive themes will be explored by KM and refined with the qualitative team. Themes will also be checked with the participant advisory group. While we will initially analyse year 1 data thematically, as year 2 and 3 data become available, we will increasingly turn to narrative analysis. This will focus on how participants order, sequence and present their stories and ‘turning points’ in their experiences of services, their relationships and understandings of themselves”.

- It might be appropriate to include a section on Reflexivity where the researchers reflect on their own positioning in relation to the topic and participants, and how that may affect the data gleaned.

Thank you - this is an excellent point and we have added a new section on Reflexivity into the manuscript:

“In order to further ensure methodological rigour and transparency, it is important to reflect on the positioning of the qualitative team in relation to the study, and how that may affect the data gleaned. KM (she/her) is an experienced qualitative researcher who has worked extensively in sensitive areas of mental health research. TW (she/her) is a lived experience co-investigator whose current PhD intends to examine how social determinants impact on the mental health of trans and gender diverse people. EK (she/her) is an experienced child and adolescent psychiatrist and clinical researcher. BY (she/her) has extensive experience as a qualitative researcher focusing on psychological processes in healthcare and clinical research”.

- Have or will any pilot interviews be conducted?

No pilot interviews were conducted.

- “As this is a two-year prospective study, participants’ experiences and wellbeing beyond this period, and their physical/medical treatment choices, will not be ascertained.” – While I am not familiar with the scientific or logistical reasons behind this decision, it does seem a shame to rule out any possibility for further follow-up as the children age. I would encourage the authors to consider whether they might design their consent and data protection procedures so that some possibility for future follow-up is left open.

We also hope to be able to extend the study and agree it would be important to follow-up on the families involved in the study. We have adjusted the sentence to leave that possibility open: “As this is a two-year prospective study, participants’ experiences and wellbeing beyond this period, and their physical/medical treatment choices, will not be ascertained, although we intend to explore further funding opportunities that would enable us to extend the data collection and participant follow-up beyond the initial two years”.

Reviewer: 2

Dr. Laura Cariola, The University of Edinburgh

Comments to the Author:

This prospective study seeks to provide a longitudinal qualitative exploration with both children and young people and their families to better understand their experiences in relation to the UK GIDS. This is a valuable study and the authors discuss the contribution of this innovative qualitative study to the existing knowledge and understanding of the topic.

Thank you so much for these positive comments.

Ethical issues have been clearly considered and discussed. Ethics codes of conduct/standards that guide this study could be mentioned.

We absolutely agree about the importance of the ethical issues in this study, and have carefully considered this. For this reason, LOGIC-Q will be conducted in line with two sets of ethical guidance. The recommendations set out by Vincent (2018) are specifically focused on the safety and agency of transgender participants in research. The Nuffield Guidance (2015) on the ethics of including children and their families in research has also ensured the agency and appropriateness of the study’s approach from the very beginning. This has been added to the ethics section of the paper:

“LOGIC-Q will draw significantly on the recommendations set out by Vincent (26) which are specifically focused on the safety and agency of gender diverse participants in research. The LOGIC-Q team are also guided by the Nuffield Guidance (27) in working with parents and children in research studies, especially ensuring that children and young people have agency in their research participation and that the data collection methods are appropriate and flexible to their needs”.

The relationship between the researcher and participants has been also discussed, and it would be useful to mention potential issues around power imbalance in the interview situation and how this would be addressed in the research. It would be also helpful to know how the researcher would address participants’ comments during the interviews/data collection and potential implications to make changes in the research design.

This is such an important point. We hope that the clarification added to the manuscript so far in terms of the recommendations from Vincent (2018) and the Nuffield Guidance, as well as how the researcher will interact with children to ensure they feel safe and comfortable during the interview

process help to demonstrate how potential power imbalance issues will be addressed in the study. We have also added the following about parents in the Informed Consent section:

“Acknowledging that there can be a potential power imbalance in research, KM will work hard to ensure that all the families understand why the research is being conducted, what their contribution will mean to the study, the value of their participation, and how their confidentiality and anonymity will be ensured and respected. KM will initially engage with the parents, answering any questions they have and ensuring that everyone is comfortable talking to her. Every participant will be informed that they have the right to withdraw from the study at any time, without providing a reason”.

In relation to Covid-19 risks to health in face-to-face data collection, it would be helpful to have some information about risk assessment.

We agree and have added the following detail:

“In light of COVID-19 and the travel and social distancing restrictions that have been in place in the UK since March 2020, all interviews will be conducted via video call or telephone. In instances when travel is not an option, interviews will be conducted via a video call or the telephone. As restrictions begin to ease, steps to monitor and ensure the safety of the families and interviewer will take precedence. In-person interviews will be undertaken on a case by case basis, where both the family and interviewer are in agreement, and travel and social contact is both safe and allowed”.

In relation to the analytic section, it is not clear how the analysis would be operationalised. The proposal of an interpretivist and pluralistic approach to thematic analysis and narrative inquiry is confusing. Arguably most qualitative approaches assume an interpretivist position. To improve the analysis section I would suggest providing a more clear outline of thematic analysis and how it will be used longitudinally from a narrative inquiry perspective. Currently, the analytic approach is unclear, for example, how the data will be analysed or coded, and how validity of the themes will be established. In case several researchers code the data, it would be important to explain how coding consistency will be established.

Thank you for these important queries. We hope these are partly answered above when further clarifying the analytical process. We would also like to add that pluralism is compatible with interpretive enquiry. We refer to it to signal our wish to avoid constraining the analyses to a specific 'brand' qualitative analysis (Chamberlain, 2000), and to reflect recommendations for qualitative researchers to innovate and adjust their methods to achieve 'fidelity' to the data and 'utility' for particular research questions (Levitt et. al., 2017). Combining two or more analytical frameworks can produce richer understandings that better reflect the complexity of human experience than drawing on a single framework (Clarke et. al., 2015). Both thematic and narrative analytical frameworks have been successfully combined previously (e.g. Simons et. al., 2008) including in a previous longitudinal qualitative study of the daily lives, networks and life transitions of children and their carers in low and middle income counties (Shukla et al, 2014).

In response to the reviewer's query about thematic analysis and how it will be used longitudinally from a narrative inquiry perspective, we have added:

“While we will initially analyse year 1 data thematically, as year 2 and 3 data become available, we will increasingly turn to narrative analysis. This will focusing on how participants order, sequence and present their stories and 'turning points' in their experiences of services, their relationships and understandings of themselves.”

References:

Chamberlain K. Methodolatry and qualitative health research. *Journal of Health Psychology*. 2000; 5(3): 285-296.

Levitt HM, et al. Recommendations for designing and reviewing qualitative research in psychology: Promoting methodological integrity. *Qualitative Psychology*. 2017; 4(1):2-22.

Clarke NJ, et al. Analytical pluralism in qualitative research: A meta-study. *Qualitative Research in Psychology*. 2015; 12(2):182-201.

Simons L, et al. Shifting the focus: Sequential methods of analysis with qualitative data. *Qualitative Health Research*. 2008; 18(1): 120-132.

Shukla N, et al. Combining thematic and narrative analysis of qualitative interviews to understand children's spatialities in Andhra Pradesh, India. 2014. NOVELLA Working Paper: Narrative Research in Action.

In relation to the aims of the study, the notion of 'describing' (page 9, line 23) is not consistent with the qualitative approach (constructivist/ interpretivist position) that preliminary seeks to explore.

Thank you for highlighting this. We have changed this to 'explore and examine'.

LOGIC-Q (page 9, line 48) requires disambiguation.

Apologies for the confusion. This has now been rectified.

Reviewer: 3

Dr. Denise Medico, University of Quebec in Montreal

Comments to the Author:

Access to care for gender diverse children is an important issue, which has been punctuated by polarized perspectives in the scientific literature, and in the mass media. It is critical to develop empirical unbiased scientific data in light of the current debates on gender diverse children. The proposed research will provide researchers and health care providers alike with significant insight. The research objectives of the study are clear, and its design method is relevant.

Thank you for these supportive comments.

A literature review would not be complete without an appraisal of worldwide initiatives. A similar yet non-longitudinal study has been done in Canada by a nationwide team of researchers led by Professor Annie Pullen Sansfaçon. The results of the aforementioned studies have been published. Another international longitudinal multi-site study (CA, USA, CH, AUS, UK) will begin in the near future.

See:

1. Pullen Sansfaçon, A., Suerich-Gulick, F., Temple-Newhook, J., Feder, S., Lawson, M., Ducharme, J., Ghosh, S., & Holmes, C. (2019). The experiences of gender diverse and trans children and youth considering and initiating medical interventions in Canadian gender-affirming specialty clinics. *International Journal of Transgenderism*, 20(4), 371-387.

<https://doi.org/10.1080/15532739.2019.1652129>

2. Pullen Sansfaçon, A., Hébert, W., Lee, E. O. J., Faddoul, M., Tourki, D., & Bellot, C. (2018).

Digging beneath the surface: Results from stage one of a qualitative analysis of factors influencing the well-being of trans youth in Quebec. *International Journal of Transgenderism*, 19(2), 184–202.

<https://doi.org/10.1080/15532739.2018.1446066>

Thank you so much for directing us to these excellent papers. We have referenced these papers on page 6: "This study also seeks to extend on the work undertaken by Pullen Sansfaçon and colleagues in Canada (11, 12). While not longitudinal, their findings on the experiences of gender diverse and trans CYP, particularly in terms of resilience and access to healthcare and the ways in which these are interwoven, highlights the vital nature of hearing the voices of lived experience. Support for CYP

needs to not only encompass their physical and mental healthcare, but also ensure they are supported through their family and social connections”.

Thank you again for your helpful review of our manuscript.

VERSION 2 – REVIEW

REVIEWER	Cariola, Laura The University of Edinburgh
REVIEW RETURNED	11-Sep-2021
GENERAL COMMENTS	Excellent. I am looking forward to seeing this published.